# Impacts of Built Environment on Risk of Women’s Lung Cancer: A Case Study of China

**DOI:** 10.3390/ijerph19127157

**Published:** 2022-06-10

**Authors:** Hongjie Xie, Rui Shao, Yiping Yang, Ramio Cruz, Xilin Zhou

**Affiliations:** 1School of Civil Engineering and Architecture, Wuhan University of Technology, Wuhan 430070, China; ssrui@whut.edu.cn (R.S.); arch.ramiro@gmail.com (R.C.); zhou.xilin@whut.edu.cn (X.Z.); 2Wuhan Branch of Chinese Center for Disease Control and Prevention, Wuhan 430010, China; xyzbb@126.com

**Keywords:** built environment, impact factors, lung cancer incidence, exploratory regression analysis

## Abstract

Built environment factors such as air pollution are associated with the risk of respiratory disease, but few studies have carried out profound investigation. We aimed to evaluate the association between the built environment and Chinese women’s lung cancer incidence data from the *China Cancer Registry Annual Report 2017*, which covered 345,711,600 people and 449 qualified cancer registries in mainland China. The air quality indicator (PM2.5) and other built environment data are obtained from the *China Statistical Yearbook* and other official approved materials. An exploratory regression tool is applied by using Chinese women’s lung cancer incidence data (Segi population) as the dependent variable, PM2.5 index and other built environment factors as the independent variables. An apparent clustering region with a high incidence of women’s lung cancer was discovered, including regions surrounding Bohai bay and the three Chinese northeastern provinces, Heilongjiang, Liaoning and Inner Mongolia. Besides air quality, built environment factors were found to have a weak but clear impact on lung cancer incidence. Land-use intensity and the greening coverage ratio were positive, and the urbanization rate and population density were negatively correlated with lung cancer incidence. The role of green spaces in Chinese women’s lung cancer incidence has not been proven.

## 1. Background

Lung cancer is a complex etiology process with numerous causative factors. Existing studies show that the main risk factors for lung cancer come from lifestyle habits (e.g., smoking, sedentary, etc.) and genetics, as well as air quality. According to the latest research, the smoking prevalence in China is 50.5% for men and 2.1% for women [1]. Compared to previous findings, women’s smoking prevalence has been stable and showing a slow decline, from 2.7% in 2013 [2] to 2.1% in 2018. It is well known that the smoking rate among Chinese women is much lower than in most countries, especially developed countries [3]. According to a retrospective proportional mortality study published in the *British Medical Journal*, 61% of women whose death is caused by lung cancer are non-smokers [4]. The decrease in Chinese women’s smoking rate is accompanied by an increase in lung cancer. In addition, secondhand smoke is not a major factor in lung cancer in women either. Only 11% of lung cancer cases among non-smoking women in China are clearly related to secondhand smoke from husbands and workplaces [5]. This suggests that risk factors besides smoking may exist.

In recent decades, many environment-related studies have identified air pollution as a significant risk factor for respiratory disease [6,7,8]. The International Agency for Research on Cancer (IARC) states that outdoor air pollution was a carcinogen in 2014 [9]. Other than that, an American Cancer Society (ACS) study noted that every 10 μg/m^3^ increase in fine airborne particles is associated with an 8% increase in lung cancer mortality [10]. ESCAPE (European study of cohorts for air pollution effects) also found the risk of all types of lung cancer increased by 22% with a 10 μg/m^3^ increase in PM10 and lung adenocarcinoma by 18% with a 5 μg/m^3^ increase in PM2.5 [11,12]. In a cross-sectional COVID-19 study, Qeadan F. et al. even found that higher average daily PM2.5 exposure is positively associated with higher county-level COVID-19 mortality rates based on data of 3142 counties across the U.S. [13]. This is not the only cause; a meta-analysis for the risk factors of Chinese lung cancer rate from 2006 to 2016 revealed air pollutants, history of respiratory disease, smoking and alcohol abuse are the most important risk factors [14].

The report ‘Evidence Review on the Spatial Determinants of Health in Urban Settings’ issued by WHO summarizes the main urban components that impact the determinants of health as land-use pattern, transport, green spaces and urban design [15]. Furthermore, some studies have demonstrated that built environments substantially impact air quality and further impact respiratory diseases [16]. Urban planning layout [17], land use [18], road traffic [19], green space [20] and other spatial elements affect the concentration and dispersion of air pollutants [21]. Wang and Chen’s study confirmed that population density, land use, road and transport, and urban planning were the main factors related to the spatial distribution of air pollution in Hangzhou [22]. Dong and Kang reported the same opinion that the spatial layout and built environment greatly influence the air quality and rural environment versus urban as a protective factor for lung cancer [23]. Several other studies have confirmed that urban planning can impact the spatial transmission pathways of airborne disease besides air pollutants [24,25,26]. In addition, other built environmental factors such as land use and density make people physically active or inactive. Physical activity has long been recognized as an effective way to reduce the risk of lung cancer. People who engage in 6–8 h of moderate physical activity per week have a significantly lower risk of lung cancer [27]. In another study, physical activity was shown to reduce the risk of lung cancer in women who have not smoked or who had smoked [28].

Wang et al. proposed a theoretical framework that includes four categories of urban planning elements similar to WHO’s [29]. Moreover, socioeconomic status also impacts health [30,31]; in this research, income is considered an independent variable. Referring to these well-established theoretical frameworks, we propose an analytical framework in which four urban planning components and social-economic statuses were broken into eight measurable indicators, as in Table 1.

However, most existing studies tend to focus on the health effect of one or a few factors. Very few studies have looked at the compound health effects and dose–response relationship of these built-environment factors, and even fewer have considered spatial factors. This research aimed to evaluate the influence of the built environmental elements on human health from the incidence of lung cancer among Chinese women. The following two questions will be answered to achieve the goal: (1) Do built environment factors affect women’s lung cancer incidence on a macro level across China? (2) What influences the direction and significance of each built environment factor on Chinese women’s incidence of lung cancer?

## 2. Methods and Data

### 2.1. Statistical Analysis

The analysis model relates to the theoretical framework in Table 1. To minimize the interference of individual behavioral factors on lung cancer, e.g., smoking and alcohol abuse, the incidence of lung cancer among Chinese women was set as the unique dependent variable. Land-use intensity (LUI), road area per capita and number of buses/10,000 persons (as one of the representations of public transportation accessibility) and greening coverage of the built-up area were selected as the independent variables corresponding to the categories of land-use pattern, transport and green space components. As we all know, urban design is somewhat a subjective variable. Urbanization rate and population density are substituted as more objective variables to avoid urban design’s subjective bias and fallacy. The air quality index (AQI), smog and dust emissions were selected to characterize air pollution variables (Figure 1).

The data were imported into ArcGIS 10.2 software and analyzed with its exploratory regression tool. The exploratory regression tool of Arcgis 10.2 evaluates all possible combinations of the input candidate explanatory variables to find the best OLS (ordinary least squares) model that explains the dependent variable based on the user-specified thresholds. The search threshold for the exploratory regression equation is shown in Table 2.

### 2.2. Study of Population

For this full-data, cross-sectional analysis, we obtained incidence data of lung cancer in Chinese women from the China Cancer Registry Annual Report 2017 [32], including cancer incidence and mortality from 449 qualified cancer registries (160 cities and 389 county-level cities) covering 345,711,600 people, which is under the framework of PBCRs. Population-based cancer registries (PBCRs) are essential in China’s national cancer control program to provide crucial cancer incidence, survival and mortality data. PBCRs have been operating for about 60 years and cover 31 provinces in mainland China; they increase along with development at different levels [33].

### 2.3. Incidence of Lung Cancer in Chinese Women

However, the scale and coverage of most cancer registration sites are inconsistent. Considering the complicated management divisions of China, analysis at the city or county level can easily lead to statistical fallacies (Figure 2). It is reasonable and practical to transform incidence data into a local database within cancer registries. At first, to eliminate the intervention of age, raw incidence data were standardized by world population (Segi population, 1960). Then, we extracted the incidence data from each registration and carried out arithmetic averaging based on provincial population.

### 2.4. Air Pollution Indicator and Its Optimization

Air quality index (AQI) and smog and dust emissions were selected as indicators to characterize air pollution. AQI is a quantitative description of air quality that includes six significant pollutants: PM2.5, PM10, SO_2_, CO, O_3_ and NO_2_. Considering that the AQI has six variables, we take an experimental regression analysis first. It is found that there is an apparent relationship between air pollution and women’s lung cancer incidence (adjusted R^2^ = 0.412, *p* < 0.01) (Table 3). However, the regression equation also shows a strong multicollinearity. The variance inflation factor (VIF) test is applied to eliminate the collinearity of variables. Any variable’s VIF ≥ 7.5 should be carefully evaluated or excluded from the regression analysis. PM10 shows a strong collinearity with PM2.5 and the significances of the remaining four variables were all greater than 0.05. After careful consideration, PM2.5 is left as the only indicator in AQI. It is also well known for its hazards to respiratory health and its capability to penetrate deep into the alveolus [34].

Considering that air quality tends to stabilize quickly and the registration time lags behind the actual time, the PM2.5 data in ‘PM2.5 Ranking of 366 Cities in China 2015’ [35] were used in this study, which include a total of 133,436 points of data. All cities’ annual average PM2.5 concentrations were used as the local PM2.5 indicators.

Smog and dust emission data were selected from the *China Statistical Yearbook* on Environment 2014 [36], which includes environmental statistics for each province according to the National Bureau of Statics.

### 2.5. Exposure Assessment: Built Environment Factors

Land-use intensity (LUI), greening coverage ratio, buses per 10,000 persons, road area per capita, urbanization rate and population density are present in the regression model as built environment factors. The LUI data are from the Land Use Evaluation Ranking of Provinces and Municipalities 2018 [37]. Data on the greening coverage ratio, buses per 10,000 persons, road area per capita and population density were sourced from the China Urban Statistical Yearbook 2015 [38]. Data on urbanization rate and per capita income per province are from publicly available data on the China National Bureau of Statistics website. Data of industrial smog and dust emissions, income per capita, and population density are too high. They differ from other indicators by orders of magnitude, so they are normalized by taking logarithms separately.

## 3. Results

### 3.1. Statistics of the Incidence of Lung Cancer in Chinese Women

After reconstructing the statistical model and computational analysis, it was found that the national average incidence of lung cancer is 23.43 per 100,000 for women. The incidence of women’s lung cancer was found to be 0.0001 at the minimum (Tibet) and 53.10 per 100,000 at the maximum (Heilongjiang), with a median value of 22.18 and a standard deviation of 9.08 (Table 4). The three provinces in China with the highest incidence of women’s lung cancer are Heilongjiang, Liaoning and Inner Mongolia. Meanwhile, the lowest three provinces are Tibet, Xinjiang and Hainan (Figure 3).

After importing data into *Arcgis 10.2* for visualization, a clustering region with a high incidence of women’s lung cancer was discovered in northeast China, including Heilongjiang, Liaoning and Inner Mongolia. Another noteworthy point is that the Shandong Peninsula and its surroundings, Jilin and Hebei, form the second-highest region surrounding Bohai Bay. They form a contiguous high incidence area around Beijing (excluding Beijing). The result of Global Moran’s I (0.139, *z* = 2.33, *p* < 0.02) also confirms this (Figure 4a). Besides this, we can find two clustering regions. One is a high-value clustering region including northeast China (Heilongjiang, Liaoning and Jilin), adjacent to the high-value areas Inner Mongolia and Hebei. The other is a low-value clustering region consisting of Qinghai and Xinjiang, surrounded by low-value areas in the northwest of China. However, no outliers are shown in the rest of China. So the three northeastern provinces combined create a region with a high incidence of lung cancer in women and needs more attention (Figure 4b).

### 3.2. Exploratory Regression Analysis

After calculation, the result of exploratory regression tools showed that 466 models passed the maximum variance inflation factor test, 409 models passed the Jarque–Bera test, 251 models passed the adjusted R-squared threshold and 21 models passed the overall spatial auto-correlation test (Moran’s I test). Still, only 4 models passed the model significance test (95% confidencial rate). Lastly, only 2 models passed all the set thresholds. Table 5 summarizes the 2 models that passed the search thresholds listed in Table 2.

The adjusted R^2^ coefficient values of model 2 reached 0.56, which was much larger than that of model 1 (0.44). Moreover, the Akaike information criterion (AIC) value (392.52) of model 2 was less than that of model 1 (395.45). Therefore, the explanatory effect of model 2 is better than model 1. Model 2 included five explanatory variables: industrial smog and dust emissions (+***), urbanization rate (−***), LUI (+***), greening coverage ratio of the built-up area (+**), and population density (−**). The variables of PM2.5, the total number of buses per 10,000 persons, road area per capita and income per capita were not significant concerning the incidence rate of lung cancer in Chinese women.

## 4. Discussion

This research is one of the few empirical studies on the connection between built environments and respiratory health. An analysis framework for correlation between the built environment and lung cancer is proposed (Figure 1). We used a high-quality, cross-sectional, nation-wide, diverse dataset, covering all 31 provinces in mainland of China.

It supports the existing theoretical hypothesis that the built environment had a weak but vital impact on lung cancer. Distribution and clustering trends were detected in lung cancer incidence in Chinese women. The results also reveal a positive correlation between land-use intensity (LUI) and the incidence of lung cancer in Chinese women, which is different from the findings of previous studies [39]. High LUI (density) is considered to encourage people to choose more physical activities which are good for health [40]. One obvious fact is that the LUI in Asian cities is generally high, especially in China; its urban fabric is entirely different from that of the sparsely populated North American cities and European cities. One possible explanation is that high LUI means more buildings that occupy a large land area, squeezing out green and public spaces. High LUI also means overcrowding with high traffic flow, more noise [41] and air pollutants [42], which can cause stress-induced chronic disease [43].

The urbanization rate and population density were found to be negatively correlated with the incidence of lung cancer in Chinese women, which is not identical to the previous findings that rural areas (lower population density) are a protective factor for lung cancer [23]. This result may be related to the fact that places with high urbanization rates tend to be large and densely populated cities. In other words, medical and health care facilities are better in big cities than in rural areas.

The greening coverage ratio was positively correlated with lung cancer incidence in Chinese women. It is also somewhat different from the general perception and previous studies. One reason may be the green coverage ratio at the provincial level is an average indicator. The granularity of the calculation is relatively coarse and insufficient to reflect the differences from province to province. Beijing has the highest green coverage ratio, while Guizhou province, which has a good green space environment, is at the bottom of the list. Second, further research is needed to verify the association between green space and health outcomes and to reveal the mechanisms involved. A study in the UK (2017) found the health status in low-income suburbs even worse than the average level because the green spaces are less accessible in these areas [44]. Another study found no significant link between the amount of green space and health [45].

The indicators of buses per 10,000 persons and road area per capita were found to be insignificant. Income indicator was the least significant variable, indicating that the incidence of lung cancer in Chinese women is not associated with the socioeconomic status of individuals. Strangely, the PM2.5 indicator is not significant either, probably because its damage to the lungs is a gradual process. This requires further study with long-term panel data.

## 5. Limitation

Although our study’s covered population is large, the number of samples is only 31. The research focused on the association between built environment indicators and women’s lung cancer prevalence at a macro-level. The resolution of the research data is not detailed enough. However, the results allude to part of the truth that there is an emergence of a cluster region of female lung cancer in northeastern China. This region happens to have a high rate of smoking among women. The cross-sectional study design itself has limitations: the design did not investigate causal associations between women’s lung cancer and environmental exposure. We could not consider the effect of behavioral factors with active smoking or forced secondhand smoke among women’s lung cancer patients. To address this potential limitation, we excluded the samples of Chinese men who have a more vital willingness to smoke actively. Future studies should use accumulated follow-up data to model changes in personal health and changes in the built environment.

The risk of residual confounding exists, as in all observational data. Apart from the built environment factors, the study did not consider other environmental attributes, including medical services, security and recreation potential. We could not study how these effects might have moderated the associations.

## 6. Conclusions

Notwithstanding the mentioned limitations, this is a systematic, large-scale analysis examining the association between women’s lung cancer incidence and a range of built environment factors. The strengths of the study are owed to the use of a dataset that was high quality in terms of population size (n > 345,711,600) and diversity (449 cancer registries) and that covered the mainland in China, including urban and rural areas. The Chinese National Cancer Center data underwent substantial centralized quality control.

The results confirmed some hypotheses of existing studies. Air quality plays a vital role in lung cancer incidence. The study also confirmed a weak but clear association between built environment factors and the incidence of lung cancer and respiratory health. Land-use intensity (LUI) and greening coverage ratio were positively correlated with lung cancer incidence rate. Urbanization rate and population density were negatively correlated with the incidence of lung cancer in Chinese women. Increasing demand for urban and health topics requires closer collaboration between scholars, architects, epidemiologists and policymakers to improve public health effects.

## Figures and Tables

**Figure 1 ijerph-19-07157-f001:**
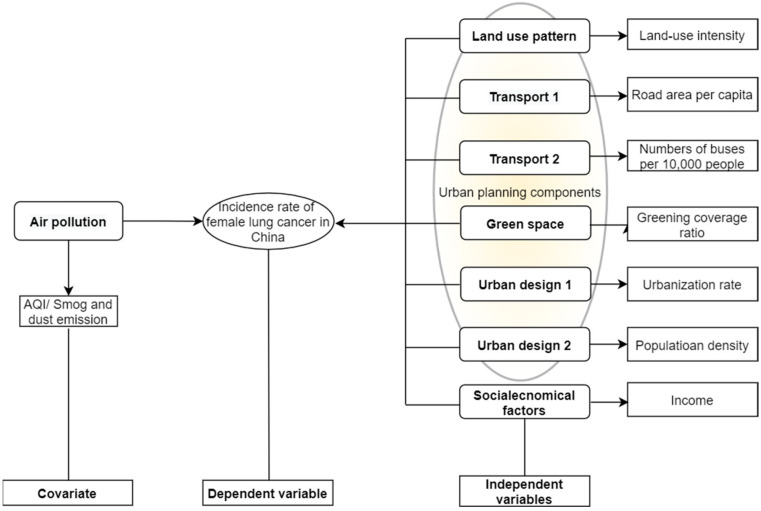
Model of women’s lung cancer incidence in China and built environment factors.

**Figure 2 ijerph-19-07157-f002:**
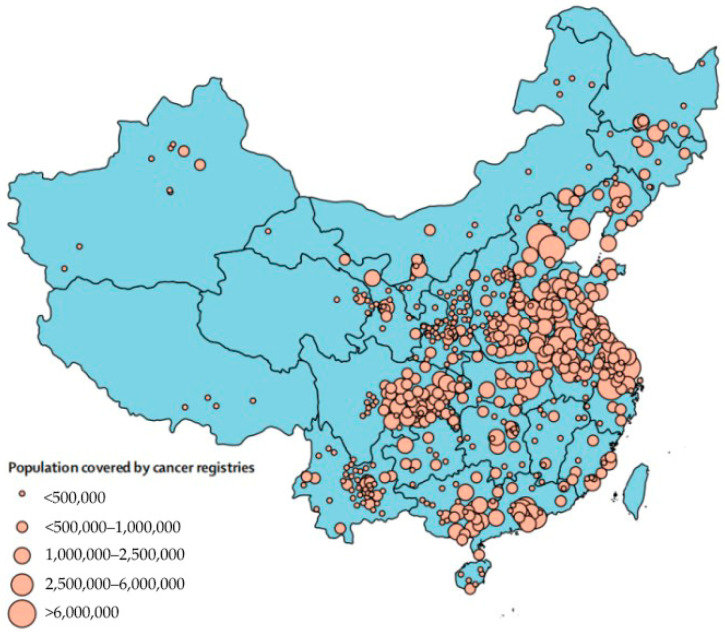
Distribution of population-based cancer registries in China (2018). Reproduced with permission from Wei W., etc., Cancer registration in China and its role in cancer prevention and control; published by The Lancet Oncology, 2020.

**Figure 3 ijerph-19-07157-f003:**
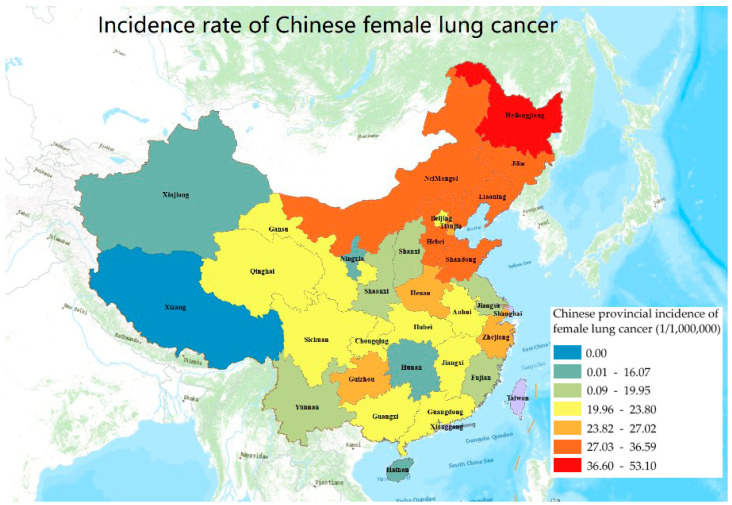
Incidence data of lung cancer in Chinese women at the provincial level.

**Figure 4 ijerph-19-07157-f004:**
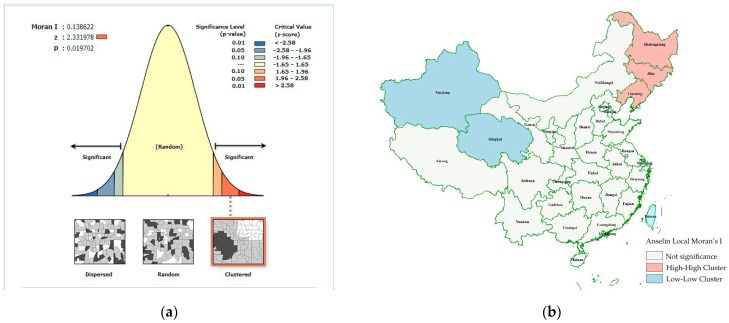
(**a**) Global Moran’s I calculation report. (**b**) Result of Anselin Local Moran’s I calculation.

**Table 1 ijerph-19-07157-t001:** Urban planning components that affect respiratory health.

Urban Planning Components	Variables	Direct Effects on Respiratory Health	Measurable Indicator
Land-use pattern	Urban planning layout	Traffic/noise/pollution	Land-use intensity
	Distribution of industrial areas	Pollutant exposure	Air quality/Smoke and dust emissions
Transport	The density of the road network	Traffic/noise/pollutant exposure	Road area per capita
	Public transport accessibility	Promotion of physical activity	Numbers of buses/10,000 persons
Green space	Green space and open space accessibility	Physical active/Recreation	Greening coverage of the built-up area
Urban design	Level of urbanization	Medical infrastructure and service levels	Urbanization rate
	Density	Risk factors for the spread of epidemics	Population density
Social-economic	Income	Health inequity	Per capita income

Source: self-created.

**Table 2 ijerph-19-07157-t002:** The threshold of the search criteria of regression tool of ArcGIS.

Search Criteria	Threshold
Minimum correction R-squared	>0.3
Jarque–Bera *p* value	>0.1
Minimum spatial autocorrelation *p*-value	>0.5

Note: A help file on exploratory regression tool box is available at https://desktop.arcgis.com/en/arcmap/latest/tools/spatial-statistics-toolbox/exploratory-regression.htm (accessed on 14 March 2022).

**Table 3 ijerph-19-07157-t003:** Experimental regression analysis.

Model	R	R^2^	Adjusted R^2^	Standard Estimate Error	Model Summary		
					Regression	Residual	Significance
1	0.741	0.549	0.412	128.513	462,727.201	379,858.260	0.005

Independent variables: (constant), NO_2_, O_3_, SO_2_, CO, PM10, PM2.5; dependent variable: incidence of women’s lung cancer (Segi population 1960).

**Table 4 ijerph-19-07157-t004:** Statistical characteristics of the incidence of lung cancer in Chinese women.

	N	Minimum	Maximum	Median	Mean	Std. Deviation
Incidence	31	0.00	53.10	22.18	23.43	9.08

**Table 5 ijerph-19-07157-t005:** Summary of models.

Model	Adj R^2^	AIC	JB	K (BP)	VIF	SA	Variables in the Model
1	0.44	395.45	0.36	0.33	2.51	0.30	−URBAN **, +LUI **, +FROG ***
2	0.56	392.52	0.67	0.22	2.79	0.50	+GREEN **, −URBAN ***, +LUI ***, −POP **, +FROG ***

Variable abbreviations: Adj R^2^: Adjusted R-squared; AIC: Akaike information criterion; JB: Jarque-Bera *p*-value K; (BP): Koenker’s studentized Breusch–Pagan statistic; VIF: Maximum variance inflation factor; SA: Global Moran’s I Variable sign (+/−); FROG: Industrial smog and dust emission; GREEN: Greening coverage ratio; URBAN: Urbanization rate; LUI: Land-use intensity; POP: Population density; Variable significance (** = 0.05, *** = 0.01).

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
