# Peer review of "Impacts of Built Environment on Risk of Women’s Lung Cancer: A Case Study of China"

_ijerph, 2022, doi:10.3390/ijerph19127157_

Round 1

Reviewer 1 Report

I accept the corrected text.

Author Response

Dear reviewers:

Thank you for your careful review. We really appreciate your efforts in reviewing our manuscript during this unprecedented and challenging time. We wish good health to you, your family, and community. Your careful review has helped to make our study clearer and more comprehensive.

Sincerely

Hongjie Xie

Reviewer 2 Report

Dear authors,

This is an ambitious and very interesting manuscript regarding an association between broadly understood built environment factors and the incidence of lung cancer and a guide to the methodology of searching for cause-and-effect relationships between biult environment factors and lung cancer incidence with simultaneous consideration of air quality.
My concerns are mostly raised by the data resolution used - mostly regarding epidemiological data and the method of their correlation with air quality data. This must be clearly discussed.
Other comments are in the main text of the manuscript (pdf version). 

Best regards

Author Response

Dear reviewer:

Thank you for your precious comments and advice. Those comments are all valuable and very helpful for revising and improving our paper, as well as the important guiding significance to our researches. We have studied comments carefully and have made correction which we hope meet with approval. Revised portion are marked in red in the paper. The main corrections in the paper and the responds to the reviewer’s comments are as flowing.

We would love to thank you for allowing us to resubmit a revised copy of the manuscript and we highly appreciate your time and consideration. Thank you very much!

Sincerely

David Hongjie Xie 

Point 1: My concerns are mostly raised by the data resolution used - mostly regarding epidemiological data and the method of their correlation with air quality data.

Response 1: Citation has been added to clarify the source of the data. The air quality indicator (PM2.5) and other built environment data are obtained from official approved materials such as the China Statistical Yearbook.

The method for this investigation cover different factors, including statistical analysis, Study of population, Incidence of China women’s lung cancer, Air pollution, and Exposure assessment: built environment factors to gather the necessary information that is used to establish the correlation between them. 

Compared to epidemiological studies, built environment and prevalence data is a fairly young field, and there is room for improvement in the granularity of research and data acquisition. In future research, I hope to conduct more in-depth research.

Point 2: Bracking space (line 15)

Response 2: Bracking space added.

Point 3: PM 2.5 concentration or for example mass loading?

Response 3: PM2.5 index.

Point 4: Space (line 30)

Response 4: Space added.

Point 5: Space (line 31)

Response 5: Space added.

Point 6: Please provide proper references.

Response 6: Proper citation has been added.

WHO. Age-standardized estimates of current tobacco use, tobacco smoking and cigarette smoking data by country. https://apps.who.int/gho/data/node.main.TOBAGESTDCURR?lang=en (accessed on 05/12).

Point 7: Space (line 38)

Response 7: Space added.

Point 8: Symbol of micro (line 42)

Response 8: Symbol of micro added.

Point 9: Symbol of micro (line 45)

Response 9: Symbol of micro added.

Point 10: Space (line 46)

Response 10: Space added.

Point 11: Space (line 49)

Response 11: Space added.

Point 12: Space (line 56)

Response 12: Space added.

Point 13: Space (line 60)

Response 13: Space added.

Point 14: Space (line 69)

Response 14: Space added.

Point 15: Planning? Or maybe between independent variables in model.

Response 15: This was a typing error. The text has been corrected and changed to: Urban planning components that affect respiratory health.

Point 16: Repeated? If not, the table should be divided with lines clearly showing which direct effect are a result of each planning element.

Response 16: The table has been adjusted and lines have been placed to show the direct effect of each indicator.

Point 17: How to differentiate industrial smog from urban one? Please discuss or change the wording from industrial to just "smog".

Response 17: Industrial fume emissions are a customary term in China's officially published statistical materials. However, to avoid ambiguity, the terminology has been changed to only “smog and dust emission”.

Point 18: Dust emissions? How to recognize this from emissions from for example secondary aerosol formation?

Response 18: Industrial fume emissions are a customary term in China's officially published statistical materials. The Air Quality Index (AQI) already considers the pollutants that are present in aerosol. This is why is not necessary to mention it again.

Point 19: Explain OLS model.

Response 19: The note has been added containing Ordinary least squares.

Point 20: Please provide references or state that these are implemented values in the model.

Response 20: A note has been added.

Note: A help file on exploratory regression tool box is available at https://desktop.arcgis.com/en/arcmap/latest/tools/spatial-statistics-toolbox/exploratory-regression.htm.

Point 21: What about people which aren't whole-life residents from Peninsula and for example half of their exposure was in completely other location? Please discuss.

Response 21: This is a cross-sectional study. Its study data are collected from a point in time or over a short time interval, so it objectively reflects the distribution of diseases at that point in time and the association between certain characteristics of people and diseases. The limitation of the cross-sectional study has been mentioned in the discussion.

Point 22: From which emissions inventory was this taken?

Response 22: Smog and dust emission is taken from The national environmental year book 2017 as noted in the paper.

Point 23: How the lung cancer incidence was correlated with air pollution (data from one year? Data from 5 years?). Where are the normality distributions tested?

Response 23: We used the annual average data of PM2.5 concentration as dependable variable to test the correlation between the lung cancer incidence and air pollution. Chinese female lung cancer incidence is a reporting data for local registries. It does not meet normality distribution rule.

Point 24: Comma (line 143)

Response 24: Comma has been added.

Point 25: Only data from 2014?

Response 25: This is a cross-sectional study. Its study data are collected from a point in time 2014. 

Point 26: With and except – I do not understand?

Response 26: The text has been changed to express the idea in a clrearer way.

“Another noteworthy point is that the Shandong Peninsula and its surrounding, Jilin, and Hebei, form the second-highest region surrounding Bohai Bay. They form a contiguous high incidence area around Beijing (except Beijing).”

The meaning of this paragraph is that Shangdong Peninsula and its surroundings, Jilin and Hebei, form a contiguous high incidence area around Beijing but Beijing itself, is not part of that high incidence area.

Point 27: Those figures 4a and 4b must be given in better resolution

Response 27: The figure’s resolution is suitable for journal publications. High resolution figure is  available.

Point 28: Acronym – Please explain.

Response 28: VIF refers to the Maximum variance inflation factor test. It has been added to the document.

Point 29: For the first time I hear about Jonquiere test - please explain how it works.

Response 29: This was a typing mistake. The name has been changed to Jarque-Bera test.

Point 30: Please explain - is a correlation coefficient that measures the overall spatial autocorrelation of the data set.

Response 30: We have added “overall” already.

Point 31: For me the greatest limitation is low resolution of data - please discuss this aspect

Response 31: In the correction we have addressed the limitation of low resolution of data, by stating:

“Although our study’s covered population is large, the number of samples is only 31. The research’s focus on the association between built environment indicators and women’s lung cancer prevalence at macro-level. The resolution of research data is not detailed enough. However, the results allude to part of the truth that the emergence of a clusters region of female lung cancer in northeastern of China. This region happens to have a high women smoking rate.”

Reviewer 3 Report

In this paper, it was studied the association between built environment factors (besides air pollution) and the risk of respiratory disease. The authors propose an analysis framework for correlation between the built environment and women’s lung cancer.

The topic is very interesting because it could provide useful supplementary elements to the normal knowledge about the relationship between the development of the pathology and the exposure to environmental pollutants, especially considering the vastness of the investigated population.

However, the results presented are summary, a bit dispersive (some are reported in the "Methods" section) and sometimes they seem not consistent (e.g. PM2.5 indicator is not significant unlike FROG=Industrial smog and dust emission).

The discussion section is too modest and not very incisive on the value added of this study to the current knowledge between the development of pathology and the factors involved. Additionally, it does not clearly highlight the results. In some cases, it appears to be contradictory (e.g.  PM2.5 indicator not significant unlike dust emission).

Despite the great interest in this subject and the effort of the authors, I believe that the work must be reorganized and it needs a more thorough approach to the issues addressed.

Further comments:

ABSTRACT

In my opinion, the abstract section should contain the number of the investigated population to highlight the scope of the study, but also describe the "main build factors" considered and which were found to have a weak but clear impact on lung cancer incidence or were found to be no significant.

Additionally, as indicated by the authors, the air quality indicator is not only PM2.5, but also Industrial smog and dust emission.

Line 13: (…) and other built environment data are obtained from approved materials.

Which?

BACKGROUND

Line 35 “With the dramatic decline in women smoking rate, this number will exceed 90% in 2018.”

This sentence doesn’t make sense. Does it refer to ref [3]?

Table 1 is not correctly formatted therefore it is not clear which “Measurable indicators” and “Direct effects on respiratory health” refer to planning elements. E.g. Land-use intensity refers to Land use pattern (as it should) or Transport? In addition, to avoid confusion, I guess it might be useful adding a column with the variables actually included in the statistical analysis of the work. In fact, Table 1 and Figure 1 seems not coherent.

Line 80. The following two questions will be answered to achieve the goal: 1)Do build environment factors affect lung cancer incidence? 2) What influences the direction and significance of each built environment factor on China women’s incidence of lung cancer?

Goal 1 does not address the aim of the article which is focused on women’s lung cancer.

METHODS

In my opinion, Table 3 should be in the results section.

Line 144. Paragraph 2.4: “Industrial smog emission data was selected from the 2014 China Statistical Yearbook on Environment”

I think it is useful to point out the main elements of this book.

RESULTS

3.1 Statistical characteristics of the incidence of China women’s lung cancer and 3.2 Spatial distribution of the incidence of women’s lung cancer in China:

The two separate paragraphs create confusion and are dispersive because they are not so different elements. Therefore, they should be merged.

Line 163: The three provinces in China with the highest incidence of women’s lung cancer are Heilongjiang, Liaoning, and Inner Mongolia. Tibet, Xinjiang, and Hainan (Figure 3).

This last sentence is meaningless and should be clarified. In addition,

DISCUSSION AND CONCLUSION

The results also reveal a positive correlation between the Land-use intensity (LUI) and the incidence of China women’s lung cancer, which is different from the findings of previous studies.

In which way? Plese, specify better.

Line 218. High-density is considered to encourage people to choose more physical 218 activities which are good for health.

This result is not reported in the results section. Is there any literature research that supports this statement?

The urbanization rate and population density were found to be negatively correlated with the incidence of China women’s lung cancer, which is not identical to the previous findings [21]. But it is consistent with the positive association between LUI and lung cancer incidence as above. This result may be related to the fact that medical and health care facilities are better in big cities than in rural areas.

In my opinion, this paragraph is not clear enough.

Line 230. The greening coverage ratio was positively correlated with China women’s lung cancer incidence. It is also somewhat different from the general perception and previous studies. One reason may be the association between green space and health outcomes is not 232 significant enough.

In my opinion, this conclusion is too weak.

Line 243: Strangely, the PM2.5 indicator is not significant either, probably because its damage to the lungs is a gradual process …

This sentence seems to contradict the positive correlation of dust emission and industrial smog.

Author Response

Dear Editors and reviewers:

Thank you for your precious comments and advice. Those comments are all valuable and very helpful for revising and improving our paper, as well as the important guiding significance to our research. We have studied comments carefully and have made correction which we hope meet with approval. Revised portion are marked in red in the paper. The main corrections in the paper and the responds to the reviewer’s comments are as flowing.

We would love to thank you for allowing us to resubmit a revised copy of the manuscript and we highly appreciate your time and consideration.

Thank you for your careful review. We really appreciate your efforts in reviewing our manuscript during this unprecedented and challenging time. We wish good health to you, your family, and community. Your careful review has helped to make our study clearer and more comprehensive.

Sincerely 

David Hongjie Xie 

Point 1: Line 13: (…) and other built environment data are obtained from approved materials.

Which?

Response 1: Proper reference has been added.

“We aimed to evaluate the association between the built environment and Chinese women’s lung cancer incidence data from China Cancer Registry Annual Report 2017 which covered 345,711,600 people and 449 qualified cancer registries in the mainland, China. The air quality indicator (PM2.5) and other built environment data are obtained from China Statistical Yearbook and other official approved materials.”

Point 2: Line 35 “With the dramatic decline in women smoking rate, this number will exceed 90% in 2018.”

This sentence doesn’t make sense. Does it refer to ref [3]?

Response 2: The text has been modified.

“The decrease in Chinese women’s smoking rate is accompanied by an increment of lung cancer. In addition, secondhand smoke is not a major factor in lung cancer in women either. Only 11% of lung cancers among non-smoking women in China are clearly related to secondhand smoke from husbands and workplaces.”

Point 3: Table 1 is not correctly formatted therefore it is not clear which “Measurable indicators” and “Direct effects on respiratory health” refer to planning elements. E.g. Land-use intensity refers to Land use pattern (as it should) or Transport? In addition, to avoid confusion, I guess it might be useful adding a column with the variables actually included in the statistical analysis of the work. In fact, Table 1 and Figure 1 seems not coherent.

Response 3: Table 1 has been modified to show the information in a clearer way. Table 1 now include urban planning components, variables, direct effect on respiratoru health of each variable, and the measurable indicator; all the changes have been made to better link Table 1 with Figure 1 and to strengthen the research.

Point 4: Line 80. The following two questions will be answered to achieve the goal: 1)Do build environment factors affect lung cancer incidence? 2) What influences the direction and significance of each built environment factor on China women’s incidence of lung cancer?

Goal 1 does not address the aim of the article which is focused on women’s lung cancer.

Response 4: The text has been edited to address the aim of the article.

“The following two questions will be answered to achieve the goal: 1)Do build envi-ronment factors affect women’s lung cancer incidence on a macro level across China? 2) What influences the direction and significance of each built environment factor on Chinese women’s incidence of lung cancer?”

Point 5: In my opinion, Table 3 should be in the results section.

Response 5: We consider this table an important part of the Methods and data section in the research. Moving this table to the results section, could leave an empty space in this part of the research, making hard to understand the general process.

Point 6: Line 144. Paragraph 2.4: “Industrial smog emission data was selected from the 2014 China Statistical Yearbook on Environment”

I think it is useful to point out the main elements of this book.

Response 6: A complementary phrase has been added to address this matter.

Smog and dust emission data was selected from the China Statistical Yearbook on Environment 2014 [35], which includes environmental statistics for each province, by National bureau of statics.

Point 7: 3.1 Statistical characteristics of the incidence of China women’s lung cancer and 3.2 Spatial distribution of the incidence of women’s lung cancer in China:

The two separate paragraphs create confusion and are dispersive because they are not so different elements. Therefore, they should be merged.

Response 7: Both paragraphs have been merged into a single one.

Point 8: Line 163: The three provinces in China with the highest incidence of women’s lung cancer are Heilongjiang, Liaoning, and Inner Mongolia. Tibet, Xinjiang, and Hainan (Figure 3).

This last sentence is meaningless and should be clarified. In addition,

Response 8: There was some information missing about Tibet, Xinjianng, and Hainan provinces.

“The three provinces in China with the highest incidence of women’s lung cancer are Heilongjiang, Liaoning, and Inner Mongolia. Meanwhile, the lowest three provinces are Tibet, Xinjiang, and Hainan (Figure 3).”

Point 9: The results also reveal a positive correlation between the Land-use intensity (LUI) and the incidence of China women’s lung cancer, which is different from the findings of previous studies.

In which way? Please, specify better.

Response 9: Not reference 21. We have added the reference [39],[40] and interpretation.

Point 10: Line 218. High-density is considered to encourage people to choose more physical activities which are good for health.

This result is not reported in the results section. Is there any literature research that supports this statement?

Response 10: Proper references have been added.

  1. Stevenson, M.; Thompson, J.; de Sá, T.H.; Ewing, R.; Mohan, D.; McClure, R.; Roberts, I.; Tiwari, G.; Giles-Corti, B.; Sun, X., et al. Land use, transport, and population health: estimating the health benefits of compact cities. The Lancet 2016, 388, 2925-2935, doi:10.1016/S0140-6736(16)30067-8.
  2. Takemi, S.; Mohammad Javad, K.; Suzanne, M.; Neville, O. Activity-Friendly built environment attributes and adult adipos-ity. Current Obesity Reports, 2014, 3, 183. doi:10.1007/s13679-014-0096-9.

Point 11: The urbanization rate and population density were found to be negatively correlated with the incidence of China women’s lung cancer, which is not identical to the previous findings [21]. But it is consistent with the positive association between LUI and lung cancer incidence as above. This result may be related to the fact that medical and health care facilities are better in big cities than in rural areas.

In my opinion, this paragraph is not clear enough.

Response 11: Important and related information has been added to the paragraph to complete the information.

The urbanization rate and population density were found to be negatively correlated with the incidence of China women’s lung cancer, which is not identical to the previous findings where rural areas (lower population density) are a protective factor for lung cancer. [22]. This result may be related to the fact that places with high urbanization rates tend to be large and densely populated cities. In other words, medical and health care facilities are better in big cities than in rural areas [42].

Point 12: Line 230. The greening coverage ratio was positively correlated with China women’s lung cancer incidence. It is also somewhat different from the general perception and previous studies. One reason may be the association between green space and health outcomes is not significant enough.

In my opinion, this conclusion is too weak.

Response 12: The paragraph has been complemented.

The greening coverage ratio was positively correlated with China women’s lung cancer incidence. It is also somewhat different from the general perception and previous studies. One reason may be the green coverage ratio at a provincial level is an average indicator. The granularity of the calculation is relatively coarse and insufficient to reflect the differences from province to province.

Point 13: Line 243: Strangely, the PM2.5 indicator is not significant either, probably because its damage to the lungs is a gradual process …

This sentence seems to contradict the positive correlation of dust emission and industrial smog.

Response 13: I'm sorry the results are such that the variables are not statistically significant enough, but that doesn't mean PM2.5 has no impact on women’s prevalence on lung cancer.

Round 2

Reviewer 2 Report

Accept in present form

Reviewer 3 Report

The authors replied in detail to all questions raised and the manuscript results significantly improved. In my opinion, the work is now suitable for publication.

This manuscript is a resubmission of an earlier submission. The following is a list of the peer review reports and author responses from that submission.

Round 1

Reviewer 1 Report

You can undeline that there are other factors  favoring the development of lung cancer, suc as family inheritance, Lynch syndrome, active or passive smoking. 

Could you refer data retrived from a control group?

- The main question is about possible link between  lung cancer in female and pollution: an an interesting topic. 
- It is an original conribution, in term of the great  number of cases observed and the particular geographic conditions
- The paper is well written and easy to read
- The conclusions are  consistent with the evidence and arguments presented, and  address the main question posed. 

Reviewer 2 Report

This is an original paper addressing an important question. However the paper has serious flaws:

A full description of the dataset used in the analysis is missing (incidence rates by province, distribution of the independent variables used in the analysis...)

The methodology is confusing. In particular, I do not understand why the authors used 466 regression models selected on the basis of statistical tests that were not justified. The abundant use of abbreviations (+LUI, -POP...) makes things even more confusing.

The authors rule out the role of smoking because of the low prevalence of smoking among Chinese women. However, variations in smoking by province of residence could also explain at least some of the variation in lung cancer incidence. This aspect should be taken into account.

The discussion does not clearly highlight the results. 

References are not cited correctly (e.g. Lancet 2019, line 50)

Reviewer 3 Report

Dear Authors, 

Thank you for the opportunity to review the paper entitled “Impacts of Built Environment on Risk of Female Lung cancer: A Case Study of China”.  The contents of the manuscript are very interesting and timely. However, the scientific soundness is low. The authors should make a little more effort in improving the structure of the manuscript- the layout of the paper is quite chaotic, as well as the English must be improved. 

Major:

I have a major issue with data acquisition, it was not clearly defined: who? from what sources? at what time? Data collection must be explained.

Minor:

  • Wrong style of reference was used
  • Lines 48-50 I suggest a sentence stylistic change to avoid "some", "some"....
  • The purpose of the study in the introduction was not precisely stated (Line 117, please identify the question as a research question
  • I suggest you remove the numbering of chapter 2 and include it in the introduction
  • Line 130 please explain the abbreviation AQI- it is only explained in line 159
  • I am also surprised by the division of chapter 3. There is no chapter on statistical analysis, characteristics of the subjects.
  • Lines 232-234 appear blank- generally the manuscript needs editing work to make it look visually better.
  • I also miss the proper discussion, it seems very modest. There are references to urbanization factors and a sudden jump to physical activity (1 sentence) I do not understand this solution. The relationship between lung cancer occurrence and physical activity is well known. Also, physical activity programs have proven to be a significant treatment factor for patients-I think it's worth mentioning in a few sentences. Please see i.e. Jastrzebski et al. Pulmonary Rehabilitation with a Stabilometric Platform after Thoracic Surgery. J Hum Kinet 2018 Dec 31;65:79-87
  • Please also specify the limitations of the study.